# Hypothesis-Driven Theory-of-Mind Reasoning for Large Language Models

**Hyunwoo Kim**[1]    **Melanie Sclar**[2]    **Tan Zhi-Xuan**[3]    **Lance Ying**[3,4]
**Sydney Levine**[3,4]    **Yang Liu**[5]    **Joshua B. Tenenbaum**[3]    **Yejin Choi**[6]

[1]NVIDIA    [2]University of Washington    [3]MIT
[4]Harvard University    [5]Amazon    [6]Stanford University

## Abstract

Existing LLM reasoning methods have shown impressive capabilities across various tasks, such as solving math and coding problems. However, applying these methods to scenarios without ground-truth answers or rule-based verification methods—such as tracking the mental states of an agent—remains challenging. Inspired by the sequential Monte Carlo algorithm, we introduce ThoughtTracing, an inference-time reasoning algorithm designed to trace the mental states of specific agents by generating hypotheses and weighting them based on observations without relying on ground-truth solutions to questions in datasets. Our algorithm is inspired by the Bayesian theory-of-mind framework, using LLMs to approximate probabilistic inference over agents' evolving mental states based on their perceptions and actions. We evaluate ThoughtTracing on diverse theory-of-mind benchmarks, demonstrating significant performance improvements compared to baseline LLMs.[1] Our experiments also reveal interesting behaviors of the recent reasoning models – e.g., o3 and R1 – on theory-of-mind, highlighting the difference of social reasoning compared to other domains.

## 1 Introduction

New reasoning algorithms and training paradigms for large language models (LLMs) have recently achieved remarkable breakthroughs in domains where well-defined ground-truth answers are readily available, enabling partial-solution evaluation and objective verification – e.g., mathematics, coding, and puzzles (Yao et al., 2023; Besta et al., 2024; OpenAI, 2024b; DeepSeek-AI, 2025b). This stands in stark contrast to social reasoning, where objective answers are not easily obtainable, making partial-solution evaluation less straightforward. Furthermore, the information-asymmetric nature of theory-of-mind (ToM) – which requires inferring hidden mental states – makes social reasoning more challenging due to its increased uncertainty (Zhi-Xuan et al., 2024b; Nafar et al., 2024).

Existing AI research on theory-of-mind has focused on either natural language benchmarks and tailored approaches to solve them (Sclar et al., 2023; Kim et al., 2023; Wilf et al., 2024; Jung et al., 2024; Jin et al., 2024), or on mental state inference in non-linguistic settings (Baker et al., 2009; Doshi & Gmytrasiewicz, 2009; Zhi-Xuan et al., 2020; Alon et al., 2023). As LLM-powered AI agents increasingly interact with humans (Collins et al., 2024), the ability to track and infer others' mental states from open-ended textual input will become crucial for broader applications and data synthesis in the social interaction domain.

To this end, we propose ThoughtTracing, an inference-time reasoning algorithm for LLMs that can infer and track the mental states of target agents. Our method is conceptually inspired by the Bayesian theory-of-mind framework (BToM; Baker et al., 2017), viewing the input context as a sequence of states and agent actions, and treating ToM reasoning as probabilistic inference about the agent's mental states based on their perceptions and actions.

---

[1]Data and code are available at https://hyunw.kim/thought-tracing

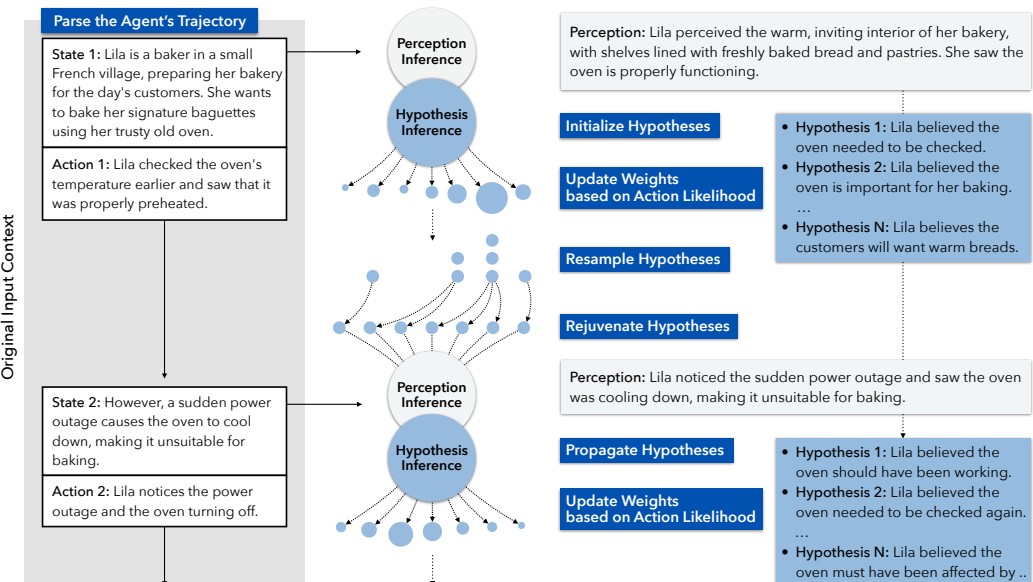

Figure 1: An illustrative overview of our ThoughtTracing algorithm. The original context is parsed into a trajectory of the target agent. At each time step, the algorithm generates multiple hypotheses about the agent's beliefs and weighs them based on the likelihood of the agent's action. These weights are then used to resample the hypotheses, prioritizing more promising ones to be used for the next time step. Further details are in Section 2.

To account for the inherent uncertainty in this task, we follow the high-level structure of sequential Monte Carlo inference (SMC; Del Moral et al., 2006; Lew et al., 2023b), tracking multiple weighted hypotheses about the agent's mental states. Importantly, however, these hypotheses are represented in open-ended natural language, and are both generated and weighted by LLMs. By using LLMs in this way, our method offers a generalizable approach to uncovering the underlying thoughts that drive agent behavior, without requiring the explicit probabilistic models assumed by BToM algorithms, and without relying on question-answer annotations or benchmark-specific assumptions.

Although ThoughtTracing is not specifically designed for solving benchmark questions, we evaluate the algorithm using state-of-the-art LLMs on four theory-of-mind benchmarks to assess whether the traced thoughts can enhance downstream question answering task performance. Experiments show that (1) ThoughtTracing consistently improves performance on all tested models across all benchmarks, (2) additional inference-time compute (e.g., chain-of-thought) further aids models to process contexts interleaved with mental states, and (3) models with ThoughtTracing outperform reasoning models (e.g., o3-mini and R1) despite using significantly shorter reasoning traces.

Furthermore, our results reveal interesting behavioral patterns in existing reasoning models on theory-of-mind tasks: (1) they do not consistently outperform vanilla LLMs using chain-of-thought reasoning, (2) they fail to generalize to similar scenarios, (3) they produce significantly longer reasoning traces for theory-of-mind questions than for factual questions, and (4) reasoning effort (e.g., output length) does not correlate with performance. These findings highlight that social reasoning differs from mathematical or programming reasoning, areas where reasoning models typically excel.

These results suggest that ThoughtTracing represents a promising step towards more robust inference-time ToM reasoning. We aim to spark new discussions on inference-time reasoning in the social domain, contrasting with the predominant focus on math and coding.

## 2 ThoughtTracing

We introduce ThoughtTracing, an inference-time algorithm that performs ToM reasoning by inferring and propagating weighted hypotheses about agents' thoughts. These hypotheses

are represented in natural language, and are generated and weighted using LLMs. Thought-Tracing follows the conceptual structure of Bayesian Theory of Mind, and takes inspiration from sequential Monte Carlo to perform approximate inference over agents' mental states.

## 2.1 Background

**Bayesian Theory of Mind** (Baker et al., 2017) frames mental state attribution as probabilistic inference over a generative model of a rational agent. It focuses on several important roles that beliefs and goals play in a theory-of-mind: the agent's perceptions and their prior beliefs jointly effect the current belief, and beliefs and goals are the causes of the agent's actions. As such, beliefs and goals can be inferred in various ways: forward simulation of beliefs from the agent's perceptions and prior beliefs, backward inference of from the agent's observed actions, or through a joint integration of all available information.

ThoughtTracing follows this framework's structure without using explicit models of rational belief updating and action selection. Instead, we use LLMs to simulate how an agent is likely to update their mental states in response to their perceptions, and to evaluate the likelihood of an agent's actions given their mental states. This enables greater generality, decomposing social inference into simpler tasks: forward simulation and likelihood evaluation by LLMs.

**Sequential Monte Carlo (SMC)** refers to a family of algorithms designed for incremental inference over sequences of posterior distributions (Del Moral et al., 2006), such as posterior inference over latent dynamics from time series data. SMC uses a collection of weighted hypotheses (called particles) to approximate each distribution in the sequence. Given particles for step $t-1$ in the sequence, SMC generates particles for step $t$ via propagation (extending each previous particle with new latent states that exist at step $t$) and reweighting (weighting each particle by the likelihood of the observed data under that particle's latent states). The particles are then resampled according to their weights (focusing samples on regions of high posterior probability) and optionally rejuvenated with Markov chain Monte Carlo (MCMC) to increase particle diversity (Chopin, 2002). To improve inference quality, SMC can make use of *custom proposals* at for propagation or rejuvenation (Lew et al., 2023a), using data-driven cues to generate more plausible hypotheses (Perov et al., 2015).

To infer hidden mental states that change over time, ThoughtTracing follows an SMC-like structure, using LLMs as proposals for propagating hypotheses about agents' mental states, and weighting these hypotheses by likelihood scores generated by LLMs. However, for simplicity, ThoughtTracing does not compute full importance weights for each particle, since this would require accessing LLM log probabilities not provided by most APIs.

## 2.2 Algorithm

ThoughtTracing leverages sequential Monte Carlo principles to maintain a dynamically updated set of natural language hypotheses about the target agent's mental state, weighting them in response to new observations. These hypotheses are updated based on the agent's perception, prior beliefs, and actions. See Algorithm 1.

**Preprocessing** Given a text input $\mathcal{C}$, we parse it into a trajectory (i.e., sequence of state-action pairs) of the target agent $\mathcal{A}$. Specifically, we use the LLM for *action labeling* of each sentence of the input. We classify each sentence whether it includes actions (e.g., physical movements or utterances) from $\mathcal{A}$. If the sentence does not include any actions from $\mathcal{A}$, it is considered a state sentence, which includes both information about the world state (descriptions of

---

**Algorithm 1** ThoughtTracing

1: **procedure** TRACE(text $c$, target agent $\mathcal{A}$)
2:     Trajectory $\tau \leftarrow$ PREPROCESS($c$)
3:     **for** $t = 1$ to $T$ **do**
4:         $(s_t, a_t, p_t) \leftarrow \tau[t]$
5:         **if** $t = 1$ **then**
6:             $H_t \leftarrow$ INITIALIZE($s_t, a_t, p_t$)
7:         **else**
8:             $H_t \leftarrow$ PROPAGATE($H_{t-1}, s_t, a_t, p_t$)
9:         **if** $a_t \neq \varnothing$ **then**
10:            $W_t \leftarrow$ UPDATEWEIGHTS($H_t, a_t$)
11:         **if** effective sample size of $H_t$ is low **then**
12:            $H_t \leftarrow$ RESAMPLE($H_t$)
13:         **if** text similarity of $H_t$ is high **then**
14:            $H_t \leftarrow$ REJUVENATE($H_t$)
15:     $\tilde{\tau} \leftarrow$ HYPOTHESESSUMMARIZATION($\tau, H$)
16:     **return** $\tilde{\tau}$       ▷ $\tau$ with traced thoughts

---

other entities or the environment) and the agent state (the agent's features or characteristics). After we obtain the state-action sequence $\{(s_t, a_t)\}_{t=1}^T$, we use the LLM for *perception inference*, producing percept descriptions $\{p_t\}_{t=1}^T$ that describe whether the agent $\mathcal{A}$ perceived each state $s_t$ and what they perceived during each action $a_t$. As a result, we obtain target agent $\mathcal{A}$'s trajectory $\tau_{\mathcal{A}} = \{(s_t, a_t, p_t)\}_{t=1}^T$.

**Initialization** ThoughtTracing begins by sampling a collection of weighted hypotheses $H_1 = \{(h_1^{(i)}, w_1^{(i)})\}_{i=1}^N$, where $h_1^{(i)}$ denotes a hypothesis and $w_1^{(i)}$ is the associated weight, from an initial distribution that approximates the step-1 posterior $p(h_1|s_1, a_1, p_1)$. In practice, the LLM is given $(s_1, a_1, p_1)$ as input, and produces a list of $N$ hypotheses, with each hypothesis assigned an initial weight of $w_1^{(i)} = \frac{1}{N}$. We set $N = 4$ for our experiments.

**Propagation** At every step $t$, each hypothesis $h_{t-1}^{(i)}$ is propagated forward to produce a new hypothesis $h_t^{(i)}$ based on the trajectory so far: $h_t^{(i)} \sim p_\theta(h_t \mid h_{t-1}^{(i)}, \{(s_\tau, a_\tau, p_\tau)\}_{\tau=1}^t)$. Here, $\theta$ denotes the parameters of an LLM and $\{(s_j, a_j, p_j)\}_{j=1}^t$ denotes the trajectory (i.e., state, action, perception) up to time $t$. In effect, the LLM is used as a data-driven proposal that generates a plausible mental state $h_t$ given the observations.

**Weight Update** After propagation, each hypothesis $h_t^{(i)}$ receives a weight update based on how well it explains the action or utterance $a_t$. The LLM is used to output the likelihood that $a_t$ would be produced under each hypothesis: $w_t^{(i)} := p_\theta(a_t|h_t^{(i)}, \{(s_j, a_j, p_j)\}_{j=1}^{t-1}, s_t, p_t)$. While it is possible to compute this likelihood from a model's log probabilities (as done in Zhi-Xuan et al. (2024b)), in practice we find that instructing the LLM to choose from six options—from "very likely (around 90%)" to "very unlikely (below 10%)"—yields better performance and stability. Moreover, most closed-source APIs (e.g., GPT-4, Gemini) do not provide access to token-level log probabilities. We map these options to numerical scores and normalize so that $\sum_i^N w_t^{(i)} = 1$.

**Resampling & Rejuvenation** Over time, some hypotheses may have very low weights, while others have very high weights. This is referred to as the degeneracy problem in SMC algorithms, and a well-known mitigation is resampling the hypotheses based on their weights (Douc & Cappé, 2005). Therefore, after the weight update, we check the effective sample size (ESS) to decide if resampling is needed. If the ESS is lower than a specific threshold, we sample $N$ new hypotheses with replacement from the existing ones $H_t$ according to their normalized weights. This discards low-weight hypotheses and duplicates higher-weight ones to obtain a new set of hypotheses. Additionally, we check the text similarity of the hypotheses to prevent them from collapsing into a single point. If similarity is over a certain threshold, we rejuvenate the hypotheses via paraphrasing.

**Hypotheses Summarization** After iterating through the entire trajectory, we aggregate the hypotheses at each time step leveraging their weights. In SMC, computing statistics such as means and variances is a common practice to summarize the distribution represented by the particles. Since our hypotheses are in natural language, we instead use the LLM to generate a weighted summary of the hypotheses set $H_t$ at time step $t$, given the past trajectory and the hypotheses along with their weights. This process can be viewed as deriving a weighted consensus over the hypotheses at each time step.

The final output of ThoughtTracing is a trajectory of the target agent $\mathcal{A}$ with the summarized hypotheses interleaved. An example is in Table 1. State, action, perception, and beliefs of $\mathcal{A}$ at each time step is concatenated as a reasoning trace formatted with HTML-style tags.

The full prompts used in ThoughtTracing can be found in the Appendix C. A case study on how the hypotheses and weights evolve over time is in the Appendix D.

---

... **<context 2>**
He then proceeds to open the third Room C cabinet and closes it as well.
**<note>**David perceived that the third Room C cabinet is empty.**</note>**
**<David's updated thoughts>**David believed that the third Room C cabinet might contain an Object E. This belief is inferred from his action of opening the cabinet to check its contents, suggesting that he thought there was a possibility of finding Object E there. However, upon opening it, he perceived that it was empty, confirming that his belief was incorrect regarding the presence of an Object E in that cabinet.**</David's updated thoughts>**
**</context 2>**
**<context 3>**
Following this, he opens and closes the stove.
**<note>**David perceived the stove in Room C, which contained an Object H and an Object J, as he opened and then closed it.**</note>**
**<David's updated thoughts>**David believed there might be an Object E in the stove. This belief is inferred from his systematic search through the storage spaces in Room C, as he had not yet found the Object E he was looking for in the cabinets. His actions of opening and closing the stove suggest that he was checking to see if it contained the desired object. However, upon opening it, he perceived that it contained an Object H and a piece of Object J, confirming that his belief regarding the presence of an Object E in the stove was incorrect.**</David's updated thoughts>**
**</context 3>** ...

---

Table 1: An example traced thoughts from our ThoughtTracing algorithm on MMToM-QA.

## 3   Experiments

Our ThoughtTracing algorithm offers a general method for tracing the mental states of a target agent, given context. While the algorithm is not explicitly designed to solve benchmark questions, we evaluate its validity by testing whether it can improve existing models' performance on four theory-of-mind (ToM) benchmarks. We apply ThoughtTracing to off-the-shelf LLMs and feed the generated thought traces along with the original context and compare their performance against other baselines, such as reasoning models.

(1) **ParaphrasedToMi** (Sclar et al., 2023) is a revised version of the classic theory-of-mind benchmark ToMi (Le et al., 2019) that addresses the limited linguistic diversity of the original ToMi by rewording all ToMi templates using GPT-3 with additional manual filtering. The resulting dataset is considerably more complex, as actions are expressed in a less straightforward way. We use a total of 600 questions and report three metrics: accuracy of (i) all questions, (ii) false belief questions, and (iii) true belief questions.

(2) **BigToM** (Gandhi et al., 2023) is a benchmark featuring synthetic narratives centered on a person's desires, actions, and beliefs, along with an event that changes the environment's state. It includes questions about the person's beliefs and actions. We find some of the narratives and ground-truth answers in BigToM include incoherent stories and errors. To address this issue, we randomly sampled 600 scenarios from the dataset and re-annotated by crowd-sourcing 510 online participants (see App. A). We only retained samples with an agreement rate higher than 90% (249 samples). We report the average accuracy of the questions.

(3) **FANToM** (Kim et al., 2023) is a multi-party conversation question-answering dataset designed to test coherent theory-of-mind capabilities. In FANToM, the speakers join and leave the conversation while it continues, making the participants hold both false and true beliefs. The benchmark includes first-order and second-order theory-of-mind questions about the beliefs of conversation participants in various formats, such as multiple-choice and list type questions. We use 64 sampled conversations from the short version of FANToM, containing a total of 1,086 questions. We report three metrics: (i) All Qs, (ii) Answerability All Qs, and (iii) Info-Accessibility All Qs, each requiring the model to correctly answer all questions for the corresponding question types for a given conversation snippet.

(4) **MMToM-QA** (Jin et al., 2024) is a multi-modal question-answering benchmark that includes questions covering the beliefs and goals of a person searching for an object (e.g., a remote controller). We use 200 questions in its text-only subset, which describes a person's activities in a household environment. Because some objects are associated with unusual

| Model | ParaphrasedToMi | | | BigToM | FANToM | | | MMToM-QA | | |
|---|---|---|---|---|---|---|---|---|---|---|
| | Avg. | False Belief | True Belief | Avg. | All Qs | Answer-ability All Qs | Info Access All Qs | Avg. | Belief | Goal |
| GPT-4o | 59.5 | 38.5 | **80.5** | 98.0 | 11.1 | 33.3 | 32.7 | 56.5 | 74.5 | 39.8 |
| GPT-4o + CoT | 67.0 | 76.0 | 58.0 | 98.0 | 40.7 | 53.7 | 71.2 | 56.0 | 82.3 | **56.1** |
| GPT-4o + TT | **76.0** | **84.5** | 67.5 | 98.8 | 41.5 | **71.7** | **76.5** | 60.0 | 78.4 | 37.8 |
| GPT-4o + TT + CoT | 74.3 | 83.0 | 65.5 | **99.2** | **54.7** | 67.9 | 74.5 | **69.0** | **93.1** | 42.9 |
| o1-preview | 68.0 | **100.0** | 36.0 | 98.3 | **44.4** | 66.7 | 63.5 | 70.4 | 96.0 | 51.0 |
| o1-low-effort | 67.8 | 98.5 | 37.0 | **99.2** | 28.3 | 41.5 | 51.0 | **76.5** | **96.1** | 57.1 |
| o1-medium-effort | 70.8 | 96.0 | 45.5 | 98.8 | 30.2 | 41.5 | 58.8 | 76.0 | 94.1 | **60.2** |
| o1-high-effort | **73.0** | 97.0 | **49.0** | 98.8 | 37.9 | 44.8 | 63.0 | **76.5** | 95.1 | 59.2 |
| o1-mini | 60.0 | 60.0 | 60.0 | **97.6** | 9.4 | **41.5** | 23.5 | 60.0 | 91.2 | 27.6 |
| o3-mini-low-effort | 64.0 | 62.5 | **65.5** | 95.2 | 0.0 | 17.0 | 11.8 | 55.0 | 71.6 | 37.8 |
| o3-mini-medium-effort | **64.5** | 90.5 | 38.5 | 97.2 | 1.9 | 22.6 | 19.6 | 69.0 | 94.2 | 42.9 |
| o3-mini-high-effort | 64.0 | **99.5** | 28.5 | 97.2 | 1.9 | 35.8 | **33.3** | **71.5** | **97.1** | **44.9** |
| DeepSeek R1 | 68.3 | 77.0 | 59.5 | 96.8 | 37.9 | 62.1 | 48.1 | 49.0 | 73.5 | 24.5 |
| Llama 3.3 70B | 57.3 | 44.5 | 70.0 | 96.4 | 0.0 | 7.4 | 0.0 | 47.0 | 50.0 | 42.9 |
| Llama 3.3 70B + CoT | 64.5 | 66.5 | 62.5 | 92.4 | 11.3 | 41.5 | 23.5 | 47.0 | 49.0 | **44.9** |
| Llama 3.3 70B + TT | 71.3 | 62.0 | **80.5** | **99.2** | 9.4 | 35.8 | 21.6 | 44.0 | 59.8 | 27.6 |
| Llama 3.3 70B + TT + CoT | **77.0** | **87.5** | 66.5 | 77.0 | **17.0** | **45.3** | **33.3** | **58.0** | **79.4** | 35.7 |
| Gemini 1.5 Pro | 54.8 | 43.5 | **66.0** | 98.0 | 1.9 | 7.5 | 2.0 | 50.0 | 70.6 | 28.6 |
| Gemini 1.5 Pro + CoT | 58.3 | 70.5 | 46.0 | 74.3 | 17.0 | 24.5 | 27.5 | 51.0 | 72.5 | 28.6 |
| Gemini 1.5 Pro + TT | **74.3** | **83.0** | 65.5 | **98.0** | 30.2 | 58.5 | 52.9 | 54.0 | 75.5 | 21.4 |
| Gemini 1.5 Pro + TT + CoT | 61.5 | 73.5 | 49.5 | 87.6 | **45.3** | **67.9** | **62.7** | **64.0** | **80.4** | **46.9** |
| Qwen 2.5 72B | 57.3 | 39.0 | 75.5 | 94.0 | 0.0 | 5.7 | 2.0 | 41.5 | 51.0 | 32.7 |
| Qwen 2.5 72B + CoT | 67.8 | 63.0 | 72.5 | 96.4 | 1.9 | 3.8 | 27.5 | 41.5 | 51.0 | 32.7 |
| Qwen 2.5 72B + TT | 74.3 | 63.0 | **85.5** | 97.2 | 1.9 | 39.6 | 21.6 | 46.0 | 60.8 | 29.6 |
| Qwen 2.5 72B + TT + CoT | **76.0** | **80.0** | 72.0 | 96.4 | **32.1** | **62.3** | **68.6** | **53.0** | **63.7** | **40.8** |
| QwQ 32B preview | 58.8 | 27.5 | 90.0 | 93.6 | 0.0 | 0.0 | 0.0 | 51.5 | 71.7 | 29.6 |

Table 2: Results across four theory-of-mind benchmarks. The underlined metric indicates the primary metric for each benchmark. The notation '+TT' denotes that our ThoughtTracing method has been applied, while '+CoT' indicates that chain-of-thought has been used. Full evaluation results can be found in Table 6 in Appendix.

locations—such as a remote controller and a fridge—contrary to commonsense, we replace room names and object labels with letters (e.g., room A and object A). We report three metrics: average accuracies of (i) all questions, (ii) belief questions, and (iii) goal questions.

**Baseline & Setup**  We apply our method to four state-of-the-art LLMs and compare their performance, and also compare with five recent reasoning models: GPT-4o (OpenAI, 2024a), o1, o1-mini (OpenAI, 2024b), o3-mini (OpenAI, 2025), DeepSeek-R1 (DeepSeek-AI, 2025a), Llama 3.3 70B Instruct (Dubey et al., 2024), Gemini 1.5 Pro (Gemini-Team, 2024), Qwen 2.5 72B (Yang et al., 2024), and QwQ 32B preview (Qwen-Team, 2024). We also compare with zero-shot chain-of-thought (CoT) reasoning (Kojima et al., 2022).

### 3.1 Overall Results

**ThoughtTracing (+TT) improves all baseline models across all benchmarks.**  Table 2 shows the results on all four benchmarks. Although ThoughtTracing does not take benchmark questions as input, its output significantly improves the model performance on the main metrics – the average accuracy (Avg.) on ParaphrasedToMi, BigToM, MMToM-QA, and the AllQs score on FANToM. For example, Llama 3.3 70B + TT, Gemini 1.5 Pro + TT, and Qwen 2.5 + TT significantly outperforms GPT-4o + CoT on ParaphrasedToMi. These results indicate that the traced mental states from ThoughtTracing provide accurate intermediate reasoning steps that helps models correctly answer mental state related questions.

**ThoughtTracing guides better reasoning trajectories.**  The fact that ThoughtTracing outperforms CoT indicates the traced thoughts exhibit better reasoning traces. Moreover, when CoT is applied on top of the traced thoughts, models' performance improves further in many cases. For example, Llama 3.3 and Qwen 2.5 achieve the first and second best scores

on ParaphrasedToMi, respectively. All models achieve the best score when ThoughtTracing and CoT is applied sequentially on MMToM-QA. Interestingly, Qwen 2.5 does not show any improvement on FANToM, when ThoughtTracing is applied. However, when CoT is further applied its performance jumps to 32.1, even outperforming o1-medium-effort and o3-mini.

On one hand, we find that, for GPT-4o and Gemini 1.5 Pro, the combination of +CoT+TT underperforms compared to +TT alone on ParaphrasedToMi. This underperformance is linked to the degradation on True Belief scenarios. In these scenarios, there is no information asymmetry—i.e., all participants share the same belief aligned with the ground-truth—making over-reasoning potentially harmful. Notably, vanilla CoT alone significantly reduces their performance on True Belief questions, suggesting that the extra reasoning steps may introduce noise or unnecessary complexity. This effect appears to be significantly stronger in closed-weight models (e.g., GPT-4o, Gemini 1.5 Pro) than open-weight ones (e.g., Llama 3.3, Qwen 2.5). Interestingly, the degradation of TT + CoT relative to TT alone is only observed in these closed-weight models, hinting at possible post-training differences.

## 3.2 Comparison with Reasoning Models

**ThoughtTracing applied models show better performance than reasoning models.** Except for o1 on MMToM-QA, models with ThoughtTracing significantly outperform o1, o1-mini, o3-mini, DeepSeek R1, and QwQ 32B preview on all benchmarks.[2] Moreover, even LLMs with CoT perform better than reasoning models, such as o3-mini, DeepSeek R1 and QwQ 32B. The o1 model also underperforms than CoT on FANToM. This suggests that reinforcing mathematical or programming reasoning does not generalize to social reasoning, necessitating separate training scheme in this domain.

**Reasoning models show performance trade-off.** A performance trade-off can be observed between false belief scenarios and true belief scenarios in ParaphrasedToMi. While o1 and o3-mini achieve near-perfect scores (i.e., 100) on the false belief scenarios, their performance on the easier true belief scenarios drops sharply to below 50 and even below 30 in some cases. A similar pattern is observed with DeepSeek R1, whose scores on the easier true belief scenarios are lower than those on the false belief scenarios. In contrast, the QwQ 32B preview follows the pattern seen in other vanilla LLMs, displaying higher scores for true belief scenarios but lower scores for false belief scenarios. Notably, QwQ significantly underperforms on FANToM, primarily due to excessive false positives. For instance, it frequently predicts that characters are aware of certain information when they are, in fact, unaware. This issue arises consistently in both list-type and binary (yes/no) questions.

Although these models do not release their training data, an interesting future direction would be to investigate how reinforcement learning influences this trade-off. On the other hand, ThoughtTracing applied models show more balanced performance across false belief and true belief scenarios, with improvements observed in both cases, except for GPT-4o.

**Output token counts and performance** Figure 2 summarizes the average output token counts for o3-mini, o1, DeepSeek R1, and GPT-4o with ThoughtTracing on ParaphrasedToMi.

(1) Output token counts in o1 and o3-mini do not correlate with performance. OpenAI's o1 and o3-mini models include an additional parameter that controls 'reasoning effort' using three settings: low, medium, and high. This parameter guides the model in determining how many reasoning tokens to generate before producing a final response. Although the o1 model with high reasoning effort outperforms the medium and low settings on ToMi, this pattern does not consistently appear across other benchmarks. For example, on BigToM, FANToM, and MMToM-QA, the o1 model with low reasoning effort achieves the highest performance. Similarly, while increased reasoning effort in o3-mini improves performance on MMToM-QA, this is not the case for other benchmarks. Moreover, performance on true belief scenarios in ParaphrasedToMi gets worse as reasoning effort increases.

---

[2]Reasoning models were provided with extra formatting prompts to align with the evaluation setup of these benchmarks. Without those formatting prompts, the scores are significantly lower.

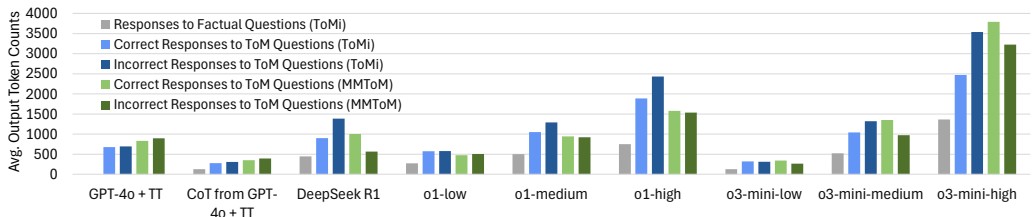

Figure 2: The average number of tokens in the reasoning trace from the reasoning models and ThoughtTracing for ParaphrasedToMi and MMToM-QA.

(2) Theory-of-mind (ToM) questions elicit more output tokens than factual questions. In ParaphrasedToMi, all reasoning models use significantly more output tokens when addressing ToM questions compared to factual ones. This suggests processing social information is more computationally demanding for these models.

(3) Token usage for incorrect and correct responses to ToM questions differs across datasets. Although a recent analysis reported incorrect responses from DeepSeek R1 and QwQ-32B preview have significantly higher token usage (Wang et al., 2025), this does not hold for ToM benchmarks. In ParaphrasedToMi, all reasoning models show significantly higher token usage for incorrect responses to ToM questions than correct ones. Moreover, the pattern is different for MMToM-QA, which the token usage is similar between correct and incorrect responses or is higher for correct ones. On the other hand, ThoughtTracing exhibits balanced token usage regardless of whether the responses to the ToM questions are correct or incorrect. We note that the relative difficulty of the questions in ToMi and MMToM-QA is consistent compared to benchmarks in other domains, and all models achieve near-perfect scores on ToMi's factual questions.

(4) Models with ThoughtTracing produce significantly shorter reasoning traces while achieving better performance. As shown in Table 2, GPT-4o+TT and GPT-4o+TT+CoT deliver the best and second-best performance among these models on ToMi, with scores of 76 and 74.5, respectively. Despite better performance, these models generate significantly fewer output tokens than o3-mini, o1 (with medium and high reasoning effort), and R1. Furthermore, compared to o1-low and o3-mini-low, GPT-4o+TT and GPT-4o+TT+CoT score almost 10 points higher. Similarly in MMToM-QA, GPT-4o+TT+CoT outperforms R1 and shows performance comparable with o3-mini-medium while using less than 50% of the tokens.

**Thought switching and performance** We also analyze the thought switching patterns for DeepSeek R1, following (Wang et al., 2025), on ParaphrasedToMi and MMToM-QA. The authors initially reported that frequent thought switching leads to underthinking and ultimately lower performance. For ParaphrasedToMi, incorrect responses included an average of 9.69 thoughts, while correct ones included an average of 6.21—consistent with the reported pattern. However, for MMToM-QA, responses for incorrect responses included an average of 6.83 thoughts similar to correct responses, which contained an average of 6.45.

### 3.3 Analysis on ThoughtTracing

**Qualitative Error Analysis** We manually evaluated ThoughtTracing's errors on ParaphrasedToMi, FANToM, and MMToM-QA. Due to the model's near-perfect performance on BigToM, we did not conduct error analysis, as the number of errors was too limited to draw meaningful conclusions.

(1) ParaphrasedToMi: We observed a recurring pattern of incorrect perception predictions. In true belief scenarios, the model sometimes incorrectly infers that the agent did not witness another agent moving an object—often due to a conservative estimation of perception (e.g., assuming the agent was not paying attention).

(2) FANToM: The model tends to overestimate the target agent's prior knowledge, occasionally assuming familiarity among characters. However, FANToM is designed such that characters are meeting for the first time, making these inferences incorrect.

These cases from ParaphrasedToMi and FANToM reflect our intentional design choice to avoid benchmark-specific assumptions—such as presuming agents always observe everything or have predefined relationships. We believe this is a principled trade-off for generalizability. Incorporating benchmark-specific cues or more explicit assumptions in the prompts will reduce these errors.

(3) MMToM-QA: We find an inherent bias across models. In many cases, when an agent searches for something and repeatedly encounters an object—despite not interacting with it—the models generate hypotheses suggesting that the object is what the agent is looking for. However, according to both social common sense and explicit models of ToM (Baker et al., 2017; Ying et al., 2025), if an agent repeatedly sees an object but continues searching elsewhere, it is likely that they are actually searching for something else. Since this error may be a side effect of perception inference in ThoughtTracing, we conduct ablation experiments.

**Ablation** We conducted ablation experiments on three models—GPT-4o, Llama 3.3 70B, and Gemini 1.5 Pro—by removing either the perception inference or the weight update (§2.2). For the perception inference removal, we directly sample hypotheses only from the state-action pairs in the trajectory at each time step. For the weight update removal, we apply uniform weights for all hypotheses across the trajectory at every time step.

| Model | Paraphrased ToMi (Avg.) | FANToM (AllQs) | MMToM (Avg.) |
|---|---|---|---|
| **GPT-4o + TT** | | | |
| Without Perception | -16.2 | -17.0 | -7.0 |
| Without Weight Update | -4.2 | 0.0 | -4.0 |
| With Self-correction | -4.2 | -3.8 | -4.0 |
| **Llama 3.3 70B + TT** | | | |
| Without Perception | -12.8 | -9.4 | 0.0 |
| Without Weight Update | -1.8 | -2.5 | -2.0 |
| With Self-correction | +0.2 | -3.7 | -1.0 |
| **Gemini 1.5 Pro + TT** | | | |
| Without Perception | -19.5 | -20.8 | -2.0 |
| Without Weight Update | -4.3 | -5.7 | -9.0 |
| With Self-correction | -5.0 | -3.8 | -9.0 |

Table 3: Score difference comparing variants of ThoughtTracing to the original algorithm on ParaphrasedToMi, FANToM, and MMToM-QA.

Table 3 shows the results. Both modifications lead to reduced performance, with the removal of perception inference causing a greater drop. This suggests not only perception inference is a crucial component in LLM's theory-of-mind reasoning (Jung et al., 2024), but also the fact that LLMs tend not to incorporate perception inference internally when generating mental state hypotheses. Additionally, this indicates the weight update based on action likelihood helps the model prioritize more promising hypotheses.

**Self-correction during Propagation** We also tested another variant of ThoughtTracing that self-corrects its previous hypothesis and predict a new hypothesis during the propagation step (§2.2). Table 3 shows the results. Although self-correction has shown a performance increase in some domains (Madaan et al., 2023; Shinn et al., 2023), it does not show better performance for hypotheses propagation on the three ToM benchmarks. In most cases, it leads to worse performance, indicating self-correction capability is not effective in our iterative updates, potentially due to the accumulation of correction errors. We suggest future works to investigate the effects of LLM's self-correcting behavior in the social domain.

# 4 Related Work

**Model-based Theory-of-Mind Reasoning** Cognitive science research has shown that humans reason about other agents' minds by assuming their actions are rational and goal-directed (Dennett, 1981; Csibra et al., 1999; Baillargeon et al., 2016). These processes have been formalized as Bayesian Theory of Mind (Baker et al., 2017; Jara-Ettinger et al., 2019), which models agents as rational actors that plan and act upon their beliefs and desires to achieve their goals. This probabilistic model can then be inverted via *inverse planning*, inferring agents' mental states from their behavior (Baker et al., 2009; Ying et al., 2023; 2025), including via SMC methods (Zhi-Xuan et al., 2020; 2024a). While ThoughtTracing is patterned on this model-based approach, we do not explicitly model rational agents, but instead use LLMs as *implicit models* of agents' behavior, and as *hypothesis generators* for agents' thoughts, enabling our method to be applied to open-ended inputs.

**Theory-of-Mind Reasoning in LLMs** Debates on whether LLMs are capable of ToM reasoning have sparked extensive controversy (Ullman, 2023; Whang, 2023; Ma et al., 2023; Shapira et al., 2024). As a result, many benchmarks for evaluating ToM reasoning have been released (Le et al., 2019; Gandhi et al., 2023; Wu et al., 2023; Shapira et al., 2023; Jin et al., 2024; Chen et al., 2024; Xu et al., 2024, inter alia) along with task complexity assessments (Huang et al., 2024). While LLMs succeed on some tasks, analyses on these benchmarks show that they are not yet at the level of human ToM, with signs of overfitting (Sclar et al., 2023) and overall poor capabilities (Sap et al., 2022; Kim et al., 2023). To mitigate this, several inference-time methods have been introduced (Sclar et al., 2023; Wilf et al., 2024; Jung et al., 2024). However, they rely on specific assumptions or few-shot examples, making them less scalable. Recently, Sclar et al. (2025) showed search for generating ToM training data and Cross et al. (2025) introduced a modular design for ToM hypothesis generation in multi-agent game scenarios. In this work, we aim to minimize assumptions and introduce a flexible algorithm capable of generating traces of a target agent's mental states in open-ended text.

**Inference-time Reasoning for LLMs** Inference-time reasoning has emerged as a pivotal area of research for LLMs, particularly in mathematics, coding, and puzzle solving. Early work, such as Chain-of-Thought (Wei et al., 2022), demonstrated the benefits of generating intermediate reasoning steps, and was later augmented by approaches capable of exploring multiple reasoning paths (Yao et al., 2023; Besta et al., 2024), aggregation across reasoning chains (Wang et al., 2023), and problem decomposition (Zhou et al., 2023). Recently, reasoning models trained via reinforcement learning—o1 (OpenAI, 2024b), R1 (DeepSeek-AI, 2025a), and QwQ (Qwen-Team, 2024)—have shown remarkable performance. Most of these approaches leverage objectively verifiable answers to enable reinforcement and accurate selection among multiple reasoning paths. However, such verification is challenging in social reasoning due to subjectivity and uncertainty. To handle uncertainty, recent methods perform probabilistic inference in LLMs via sequential Monte Carlo (Lew et al., 2023b; Zhao et al., 2024; Loula et al., 2025), but have not been used to infer and track mental states. In this work, we introduce a general SMC-like algorithm capable of operating effectively in the social domain while bypassing the need for objective verification.

## 5    Conclusion

In this paper, we introduced ThoughtTracing, a new inference-time algorithm that performs approximate inference over agents' mental states. Our approach draws inspiration from the sequential Monte Carlo (SMC) method and follows the conceptual structure of Bayesian Theory of Mind (BToM; Baker et al., 2017). ThoughtTracing leverages an SMC-like structure by using LLMs as proposals for natural language hypotheses, which are then weighted with actions from the given text—all without requiring ground-truth answers to questions in datasets. Our evaluation on four theory-of-mind benchmarks demonstrates its effectiveness, significantly improving upon baseline LLMs and outperforming recent reasoning models such as o3-mini and R1. Additionally, our findings reveal that reasoning models show different behaviors in the social domain compared to their remarkable performance in areas like math and coding. Nonetheless, there remains room for improvement due to errors in areas such as how hypotheses about agents' goals are generated (Section 3.3). This suggests that more work needs to be done to achieve the reliability that explicit BToM models demonstrate in closed-world settings (Zhi-Xuan et al., 2022; 2024b; Ying et al., 2025). We hope these insights pave the way for future research on inference-time reasoning in social settings, where AI agents will play an increasingly important role.

## Acknowledgments

This work was supported in part by the Amazon Research Gift Grant. Tan Zhi-Xuan is funded by the Open Philanthropy AI Fellowship.

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

# A    BigToM Re-annotation

The original BigToM dataset (Gandhi et al., 2023) includes 6 tasks, each with 200 stimuli. In our re-annotation, we took the first 100 stimuli from each task category.

We recruited 510 participants (Average age = 41.33, 238 males, 265 females, 7 others) via Prolific. To minimize potential bias, we did not conduct tutorials for this task. The experiment took place over a customized interface. Each participant was randomly assigned to complete 30 stimuli randomly sampled from the set of stimuli. Each stimulus received on average 13 ratings.

Table 4 shows the average agreement for each question type. We manually went over the re-annotation and found samples with an agreement rate below 90% often contained incoherent stories or debatable answer options. Therefore, we only use samples with an agreement rate above 90% in our experiments.

|  | Backward Belief (FB) | Backward Belief (TB) | Forward Action (FB) | Forward Action (TB) | Forward Belief (FB) | Forward Belief (TB) |
|---|---|---|---|---|---|---|
| Mean | 0.79 | 0.68 | 0.69 | 0.90 | 0.94 | 0.78 |
| Std Dev | 0.17 | 0.21 | 0.22 | 0.10 | 0.11 | 0.14 |

Table 4: Average agreement and standard deviation for each question type in BigToM.

# B    ThoughtTracing Setup

We use greedy decoding throughout our ThoughtTracing algorithm. We set the number of hypotheses to 4 across all experiments. The threshold for the effective sample size is set to 2. We measure the text similarity across hypotheses using the Jaccard similarity, and the threshold for rejuvenation is set to 0.25. For action labeling, we use GPT-4o-2024-08-06 for ParaphrasedToMi and BigToM, and a rule-based algorithm for FANToM and MMToM-QA. For FANToM, we cache the traced thoughts of each character in the conversation for efficiency.

# C    Input Prompts used in ThoughtTracing

For generalization across different models and benchmarks, we use simple prompts for ThoughtTracing, leaving more sophisticated prompts for future work. Note, the additional tokens introduced by ThoughtTracing are bounded by the length of the input text.

> **Perception Inference**
>
> You are an expert perception tracker tasked with determining whether {target agent} perceived the target context. Briefly describe what {target agent} saw or why {target agent} could not see the target context. Make your response concise.
>
> {context history}
>
> <target context>{current context}</target context>

> **Initialization**
>
> {context}
>
> Generate a numbered list of {n} hypotheses what were {target agent}'s thoughts (e.g., beliefs, intent) that led to the action above. Do not add any additional comments.

## Propagation

You are an expert assistant trying to predict {target agent}'s thoughts.

<previous context>
{context}
</previous context>
<previous prediction regarding {target agent}'s thoughts>
{hypothesis}
</previous prediction regarding {target agent}'s thoughts>

<current context>
{new context}
</current context>

What did {target agent} believe?

## Weight Update

Your job is to evaluate the probability of actions/utterance under a given fact. Use common sense: for instance, if someone is searching for an item, they are likely to take it once they find it rather than merely observing it. If they don't take it and just sees it, it indicates a lack of interest and that was not what they were looking for. Briefly explain the probability of the action/utterance under the given fact first and then give the answer option with prefix 'Answer:'

{context}
<{target agent}'s thoughts>
{hypothesis}
</{target agent}'s thoughts>
<next action>{action}</next action>
<note>{perception}</note>

Question: Based on the context and {target agent}'s thoughts provided, would {target agent} do the next actions or say the next utterances described above? Let's think step by step and give the final answer.
(a) Very Likely (Around 90%)
(b) Likely (Around 70%)
(c) Somewhat Likely (Around 60%)
(d) Somewhat Unlikely (Around 25%)
(e) Unlikely (Around 20%)
(f) Very Unlikely (Below 10%)

## Rejuvenation

Your task is to paraphrase the following text. Make sure to keep the meaning of the text intact while rephrasing them. Do not add any additional comments.

{hypothesis}

---

**Hypotheses Summarization**

{context}

<{target agent}'s thoughts>
Prediction 1 (Weight: {weight 1})
{hypothesis 1}

Prediction 2 (Weight: {weight 2})
{hypothesis 2}

...

Prediction N (Weight: {weight N})
{hypothesis N}
</{target agent}'s thoughts>

Question: What did {target agent} believe?

---

## D    Example Thought Tracing Process

Given the length of the interleaved hypotheses, we provide a summarized example of how the hypotheses evolve over time and how the weights are updated during the tracing process in Table 5.

---

**State 1:** ...In the Room B, there is a sofa, a cabinet, a desk, and a coffee table. ...The Room D is furnished with a sofa, a coffee table, and a desk. ...The Room C is equipped with eight cabinets, a microwave, a fridge, a dishwasher, a Room C table, and a stove. ...Jessica is situated in Room B.
**Action 1:** She proceeds towards the Room A, approaches the Room A cabinet, opens it, and subsequently closes it.
**Hypotheses:**

- **(Weight: 0.33)** Jessica was curious about the contents of the cabinet in Room A, driven by a general exploratory behavior.
- **(Weight: 0.33)** Jessica had a specific task or goal that required her to verify the contents of the cabinet in Room A, leading her to open and then close it after finding it empty.
- **(Weight: 0.20)** Jessica believed that Room A might contain items of interest or value, motivating her to investigate the cabinet there.
- **(Weight: 0.12)** Jessica intended to check if the cabinet in Room A was similar to the one in Room B, possibly looking for specific objects.

**State 2:** None
**Action 2:** She then navigates her way to the Room C, heads towards the fourth Room C cabinet, opens it, and then shuts it.
**Hypotheses:**

- **(Weight: 0.25)** Jessica was curious and exploratory; she noted the variability in cabinet contents (empty in Room A, Object K in Room C), leading her to believe contents differ across rooms.
- **(Weight: 0.25)** Jessica had a specific goal, believed cabinets might hold a desired item, and methodically searched until she found Object K in Room C, reinforcing her belief in a purposeful search.
- **(Weight: 0.25)** Jessica believed cabinets might contain valuable items, remained curious despite an empty cabinet in Room A, and was validated by finding Object K in Room C.
- **(Weight: 0.25)** Jessica expected cabinets in other rooms to contain objects like those in Room B; after an empty cabinet in Room A, she kept searching and discovered Object K in Room C, altering her expectations.

...

---

Table 5: A summarized example of the ThoughtTracing process on MMToM-QA.

The first set of hypotheses is initialized from State 1 and Action 1. We update their weights using Action 1. Since the diversity of the hypotheses is above the threshold, rejuvenation is not performed. Additionally, the effective sample size is above the threshold, so resampling is not carried out.

Next, we derive the second set of hypotheses by propagating from the first set. Consequently, the second set inherits the contents of its parent hypotheses. The likelihood of the second set is uniform, as all four previous hypotheses are approximately equally effective in predicting that Jessica is heading to Room C (i.e., Action 2).

## E   Experiment Results

Table 6 shows the full evaluation results on ParaphrasedToMi, FANToM, and MMToM-QA.

| Model | ParaphrasedToMi | | | | FANToM | | | | | MMToM-QA | | | | | | |
|---|---|---|---|---|---|---|---|---|---|---|---|---|---|---|---|---|
| | First-order False Belief | Second-order False Belief | First-order True Belief | Second-order True Belief | Belief Multi-choice | Info Accessibility List type | Info Accessibility Yes/No type | Answerability List type | Answerability Yes/No type | Type 1.1 | Type 1.2 | Type 1.3 | Type 2.1 | Type 2.2 | Type 2.3 | Type 2.4 |
| GPT-4o | 15.0 | 62.0 | 98.0 | 63.0 | 64.5 | 38.5 | 91.8 | 63.0 | 79.7 | 94.0 | 40.7 | 67.9 | 17.7 | 50.0 | 1.6 | 56.7 |
| GPT-4o + CoT | 69.0 | 83.0 | 79.0 | 37.0 | 83.9 | 78.8 | 93.7 | 74.2 | 83.4 | 100.0 | 41.2 | 81.2 | 24.1 | 64.0 | 4.2 | 55.0 |
| GPT-4o + TT | 89.0 | 80.0 | 85.0 | 50.0 | 76.1 | 82.4 | 96.2 | 83.0 | 92.4 | 100.0 | 56.2 | 56.2 | 10.3 | 76.0 | 8.3 | 55.0 |
| GPT-4o + TT + CoT | 94.0 | 72.0 | 79.0 | 52.0 | 89.1 | 80.4 | 97.3 | 77.4 | 92.9 | 100.0 | 94.1 | 84.4 | 41.4 | 68.0 | 0.0 | 65.0 |
| o1-preview | 100.0 | 100.0 | 57.0 | 15.0 | 83.9 | 69.2 | 91.9 | 81.5 | 91.4 | 94 | 98.8 | 90.1 | 35.5 | 95.0 | 3.3 | 52.2 |
| o1-low-effort | 98.0 | 99.0 | 58.0 | 16.0 | 67.4 | 52.9 | 95.1 | 66.0 | 83.0 | 94.4 | 100.0 | 82.8 | 0.0 | 72.0 | 0.0 | 80.0 |
| o1-medium-effort | 98.0 | 94.0 | 94.0 | 27.0 | 67.4 | 60.8 | 95.1 | 60.4 | 87.0 | 88.9 | 100.0 | 79.3 | 4.2 | 100.0 | 4.2 | 75.0 |
| o1-high-effort | 99.0 | 95.0 | 64.0 | 34.0 | 70.6 | 63.0 | 92.4 | 62.1 | 84.9 | 88.9 | 100.0 | 86.2 | 0.0 | 100.0 | 0.0 | 75.0 |
| o1-mini | 46.0 | 74.0 | 79.0 | 40.0 | 65.2 | 39.2 | 84.8 | 67.9 | 82.9 | 100.0 | 85.3 | 87.5 | 0.0 | 68.0 | 0.0 | 50.0 |
| o3-mini-low-effort | 55.0 | 69.0 | 78.0 | 53.0 | 34.8 | 19.6 | 80.3 | 35.8 | 71.3 | 100.0 | 44.1 | 68.8 | 20.7 | 80.0 | 0.0 | 55.0 |
| o3-mini-medium-effort | 87.0 | 94.0 | 61.0 | 16.0 | 34.8 | 23.5 | 84.6 | 43.4 | 76.7 | 100.0 | 88.2 | 93.8 | 24.1 | 92.0 | 0.0 | 60.0 |
| o3-mini-high-effort | 99.0 | 100.0 | 52.0 | 5.0 | 30.4 | 41.2 | 87.3 | 54.7 | 78.1 | 100.0 | 100.0 | 90.6 | 31.0 | 88.0 | 0.0 | 65.0 |
| DeepSeek R1 | 62.0 | 100.0 | 86.0 | 35.0 | 86.3 | 66.7 | 89.5 | 75.9 | 88.2 | 97.2 | 44.1 | 78.1 | 6.9 | 56.0 | 0.0 | 40.0 |
| Llama 33.70B | 17.0 | 72.0 | 94.0 | 46.0 | 69.9 | 0.0 | 83.5 | 18.5 | 76.1 | 100.0 | 20.6 | 25.0 | 55.2 | 56.0 | 0.0 | 60.0 |
| Llama 33.70B + CoT | 53.0 | 80.0 | 79.0 | 46.0 | 73.9 | 27.5 | 90.2 | 52.8 | 84.6 | 100.0 | 17.6 | 25.0 | 58.6 | 60.0 | 0.0 | 60.0 |
| Llama 33.70B + TT | 68.0 | 56.0 | 95.0 | 66.0 | 63.0 | 27.5 | 90.0 | 52.8 | 87.3 | 38.2 | 38.2 | 37.5 | 3.4 | 52.0 | 0.0 | 60.0 |
| Llama 33.70B + TT + CoT | 88.0 | 87.0 | 83.0 | 50.0 | 71.7 | 35.3 | 96.2 | 56.6 | 91.9 | 100.0 | 94.1 | 40.6 | 20.7 | 60.0 | 0.0 | 70.0 |
| Gemini 1.5 Pro | 26.0 | 61.0 | 83.0 | 49.0 | 71.7 | 2.0 | 81.3 | 18.9 | 79.9 | 100.0 | 47.1 | 62.5 | 0.0 | 72.0 | 0.0 | 50.0 |
| Gemini 1.5 Pro + CoT | 61.0 | 80.0 | 71.0 | 21.0 | 80.4 | 29.4 | 82.7 | 52.8 | 80.8 | 100.0 | 52.9 | 62.5 | 0.0 | 72.0 | 0.0 | 50.0 |
| Gemini 1.5 Pro + TT | 80.0 | 86.0 | 85.0 | 46.0 | 80.4 | 80.4 | 86.2 | 86.8 | 89.0 | 100.0 | 82.4 | 71.9 | 0.0 | 64.0 | 0.0 | 25.0 |
| Gemini 1.5 Pro + TT + CoT | 87.0 | 60.0 | 79.0 | 20.0 | 82.6 | 76.5 | 89.3 | 79.2 | 91.3 | 94.4 | 88.2 | 56.2 | 10.3 | 84.0 | 29.2 | 75.0 |
| Qwen 2.5 72B | 22.0 | 56.0 | 85.0 | 66.0 | 34.8 | 5.9 | 87.3 | 34.0 | 49.6 | 97.2 | 14.7 | 37.5 | 10.3 | 60.0 | 12.5 | 55.0 |
| Qwen 2.5 72B + CoT | 50.0 | 76.0 | 84.0 | 61.0 | 76.1 | 37.3 | 90.8 | 49.1 | 43.6 | 97.2 | 14.7 | 37.5 | 10.3 | 60.0 | 12.5 | 50.0 |
| Qwen 2.5 72B + TT | 65.0 | 61.0 | 91.0 | 80.0 | 48.9 | 21.6 | 95.2 | 52.8 | 84.6 | 100.0 | 47.1 | 6.9 | 0.0 | 64.0 | 8.3 | 45.0 |
| Qwen 2.5 72B + TT + CoT | 90.0 | 70.0 | 86.0 | 58.0 | 70.7 | 68.6 | 96.8 | 71.7 | 90.5 | 94.4 | 61.8 | 31.2 | 41.4 | 56.0 | 6.5 | 65.0 |
| QwQ 32B Preview | 28.0 | 27.0 | 89.0 | 91.0 | 67.4 | 0.0 | 59.5 | 18.9 | 55.9 | 97.2 | 61.8 | 34.4 | 3.4 | 52.0 | 16.7 | 55.0 |

Table 6: Full evaluation results across three theory-of-mind benchmarks. The notation '+TT' denotes that our ThoughtTracing method has been applied, while '+CoT' indicates that chain-of-thought has been used. We exclude BigToM because most of the models achieved a near-perfect score on our re-annotated subset.

