# OpenReview forum: "Hypothesis-Driven Theory-of-Mind Reasoning for Large Language Models"
_colmweb.org/COLM/2025/Conference — COLM 2025_

### Official Review · Reviewer_jvBW · 2025-05-12

**Rating:** 7
**Confidence:** 3
**Ethics Flag:** 1

**Summary:**

This work introduces 'Thought Tracing' (TT) an inference-time algorithm designed to infer and track the mental states of target agents in scenarios where ground truths or verification is not possible. Thought tracing is based on the concepts of Bayesian Theory of Mind and uses Sequential Monte Carlo for inference over agents' mental states.

Thought tracking algorithm at core:
Given input text is first parsed into a trajectory of state, actions. Then, at each time step, it generates multiple hypotheses about the agent's beliefs and weights them according to the likelihood of the agent's actions. These weights are used to resample hypotheses, prioritizing more promising ones for subsequent steps. The process involves initialization of hypotheses, propagation of hypotheses based on the agent's trajectory, weight updates based on action likelihood, and optional resampling and rejuvenation of hypotheses. Finally, a weighted summary of the hypotheses is generated at each time step

The authors also evaluated this approach on various Theory of Mind (ToM) tasks and demonstrate that providing these traces along with context significantly improves the baseline models

**Questions To Authors:**

1. What is different about ParagraphToMi that often CoT + TT is underperforming compared to TT alone?
2. How do you plan to handle the bias for hypothesis generation?

**Reasons To Accept:**

1. The approach doesn't require any predefined labels and it starts from the given input to form trajectories, all during LLM inference time, this would save a lot of annotation cost.
2. The use of LLMs to generate perception based on state action pairs seems novel and the ability to handle multiple hypotheses using weighted system (occasionally with paraphrasing) seems stable.
3. Robustness coming from hypothesis summarization and sampling seems to make this a generalizable solution than catering to a specific domain within ToM, more experiments here would make a stronger case.
4. The experiments are detailed and evaluations on all relevant reasoning models are performed, showing better performance than CoT.
5. The authors claim that with help of Thought Tracing is able to get better performance compared to other reasoning models using fewer tokens.

**Reasons To Reject:**

1. Although the solution theoretically is generalizable, the prompts are simple (authors mention this too), overall this seem to suggest that the method might need to be developed more to call it truly generalizable and robust across domains.
2. The hypothesis generation is one of the crucial components of the algorithm and more insights into that would have been helpful.
3. Some details about additional tokens needed for computing Thought Traces would have been helpful.

---

> ### Author Response · Authors · 2025-06-02
>
> Thank you for your insightful and encouraging feedback! We appreciate your recognition of our method’s scalability—operating without predefined labels—and the novelty of using LLMs to infer perceptions from state-action pairs. We’re pleased that you found the multi-hypothesis framework with weighted sampling and summarization to be stable and generalizable. We also value your acknowledgment of the approach’s robustness, the comprehensive experimental evaluations, and its improved performance and efficiency compared to CoT and other reasoning models. We address your remaining concerns below.
>
> ### Prompt Simplicity and Generalizability
> While we acknowledge that the prompts are relatively simple, we emphasize that **this design choice was intentional to isolate the effects of the ThoughtTracing algorithm itself**, and we believe that it is a strength of our method that it performs well despite using only relatively simple prompts without benchmark-specific assumptions. The method's robustness is demonstrated through consistent gains across four diverse Theory-of-Mind scenarios spanning various input formats and tasks: tracking beliefs about object locations, interpreting intentions, and modeling knowledge in narratives and conversations.
>
> ### Insights on Hypothesis Generation
> In our experiments, we observed that hypothesis diversity—particularly during initialization—is a key factor. To encourage diversity, we generate all N initial hypotheses in a list-style format in a single prompt (see Appendix C). This strategy yields significantly more varied hypotheses than sampling from the model N times independently, which tends to produce near-identical outputs due to the limited information in the first (state, action, perception) triple. As a result, it prevents early convergence and hypothesis collapse during propagation. We note that similar “sampling without replacement” strategies have been used in SMC and importance sampling to increase hypothesis diversity, as described in the literature on multiple importance sampling (see e.g. [Elvira et al (2019)](https://projecteuclid.org/journals/statistical-science/volume-34/issue-1/Generalized-Multiple-Importance-Sampling/10.1214/18-STS668.full)).
>
> ### Additional Tokens Needed for Computing Thought Traces
> The additional tokens introduced by our ThoughtTracing algorithm are bounded by the length of the character’s parsed trajectory, which is derived directly from the input text. Each iteration generates a natural language hypothesis based on the current state-action-perception tuple. Other than the input text, our method uses minimal prompts without lengthy instructions or domain-specific assumptions—helping ensure generality and efficiency. For transparency, the exact prompts are included in Appendix C. We will add this detail in the updated draft.
>
>
> ### What is different about Paraphrased-ToMi that often CoT + TT is underperforming compared to TT alone?
> Thank you for the great question! We hypothesize that the underperformance of TT + CoT compared to TT alone on Paraphrased-ToMi is related to degradation on True Belief scenarios (see Table 2). These scenarios lack information asymmetry—i.e., all participants share the same belief aligned with the ground-truth—making over-reasoning potentially harmful. Notably, vanilla CoT alone significantly reduces performance on True Belief questions, suggesting that the extra reasoning steps may introduce noise or unnecessary complexity.
>
> This effect appears to be significantly stronger in closed-source models (e.g., GPT-4o, Gemini 1.5 Pro) than open-source ones (e.g., Llama 3.3, Qwen 2.5). Interestingly, the degradation of TT + CoT relative to TT alone is only observed in these closed-source models, hinting at possible post-training differences. We will include this analysis in the updated version of our paper.
>
>
> ### How do you plan to handle hypothesis generation bias?
> Thank you for raising this important point! One promising direction we are actively considering is the incorporation of **counterfactual hypotheses** during the propagation or rejuvenation phases of ThoughtTracing. Because our hypotheses are represented in natural language, we can systematically prompt the model to generate alternative mental states by explicitly considering "what if" scenarios—e.g., what the agent might have believed if a perception or action had been different.
>
> This counterfactual augmentation can help overcome the innate biases of the base LLM by encouraging the model to explore plausible but underrepresented belief states that may not emerge through vanilla hypothesis sampling alone. For example, in object search scenarios, this would allow the model to contrast “the agent believes object X is the target” with “the agent believes object X is not the target, despite seeing it.” We believe this systematic generation of counterfactuals will increase the coverage and diversity of hypotheses and reduce overcommitment to biased priors.

---

> > ### Comment · Reviewer_jvBW · 2025-06-10
> >
> > Thank you for your responses to my questions!

---

### Official Review · Reviewer_gn2m · 2025-05-13

**Rating:** 6
**Confidence:** 3
**Ethics Flag:** 1

**Summary:**

Inspired by the sequential Monte Carlo algorithm, this paper introduces ThoughtTracing, an algorithm designed to improve the theory of mind capabilities of large language models.

**Reasons To Accept:**

- The proposed algorithm delivers strong improvements over the baselines
- The proposed algorithm is applicable to different LLMs
- Bringing Bayesian Theory-of-Mind and LLMs is interesting (and novel, to the best of my knowledge)

**Reasons To Reject:**

- In the current form, it is unclear to me whether the proposed algorithm can work beyond the Theory-of-Mind tasks. While I understand this is the focus of the paper, I think it would be beneficial if the authors could expand on how/why this is not just scaffolding for theory of mind tasks.
- The paper would benefit from comparing with stronger baselines
- Moving away from the Bayesian claim by using the 6 points assessment (very likely, likely, ..)

---

> ### Author Response · Authors · 2025-06-02
>
> Thank you for your thoughtful and constructive feedback, as well as for recognizing the strengths of our work! We're especially grateful for the acknowledgment that our algorithm delivers strong improvements over baselines, is applicable across different LLMs, and offers a novel integration of Bayesian Theory-of-Mind with language models. These points align closely with our goals, and we appreciate the recognition. Below are the responses to your helpful comments for further discussion.
>
> ### Applicability beyond Theory-of-Mind tasks
>
> We thank the reviewer for raising this important point! ThoughtTracing is not restricted to ToM benchmarks. It provides **a general inference-time approach for tracking latent mental states over time**, which is the fundamental building block of social reasoning, especially in uncertain, information-asymmetric scenarios. This capability underpins a broad range of real-world applications such as assistive agents, contextual decision-making, and multi-agent coordination.
>
> 1. For instance, the **MMToM-QA benchmark that we tested on is an adaptation of a robotics task** where the model must determine what the person is searching for in order to provide assistance. Keeping such downstream applications in mind, we tested our method across diverse tasks: tracking beliefs about object locations (MMToM-QA, ToMi), interpreting intentions (MMToM-QA, BigToM), and modeling knowledge in narratives and conversations (FANToM).
>
> 2. Additionally, we conducted **a new small experiment using the Confaide benchmark, which evaluates the contextual privacy understanding of LLMs—a task that heavily relies on social reasoning**. Specifically, we tested Qwen 2.5 on Tier 4, where the model must generate personal action items from a meeting transcript while adhering to privacy norms (i.e., keeping private information while sharing public information). The vanilla Qwen 2.5 model exhibited a high average error rate of 0.9, indicating poor performance. Notably, applying Chain-of-Thought reasoning further increased the error rate to 0.96. **However, with ThoughtTracing applied, the error rate dropped to 0.785**, suggesting that the inferred thought traces helped the model respect the privacy boundaries of the target agent.
>
> 3. Finally, we emphasize that the reasoning traces generated by ThoughtTracing can enhance the efficiency and precision of downstream reasoning models—particularly in domains where latent mental state inference is essential and where existing reasoning models continue to struggle.
>
>
> ### Additional Baseline
>
> We compared our method against OpenAI’s o3-mini, o1, gpt-4o, DeepSeek R1, Qwen 2.5, QwQ, and Llama 3.3 (Table 2), which are the most recent state-of-the-art reasoning and instruction-tuned models at the time of our submission. We believe this selection provides a strong and diverse baseline for comparison. That said, we welcome further suggestions from the reviewer on any specific baselines we may have overlooked, and would be happy to include additional comparisons in a future revision.
>
>
> ### Bayesian claim regarding 6-point likelihood scale approximation
>
> We acknowledge the reviewer’s point and will revise the draft to more clearly communicate that the underlying reasoning framework of ThoughtTracing remains **Bayesian in spirit**, while adapting to the practical constraints of working with LLMs. As described in our introduction, ThoughtTracing is **conceptually inspired by Bayesian Theory-of-Mind and sequential Monte Carlo (SMC) methods**. Empirically, we found that this approximation yields better performance and stability across models. Despite this pragmatic shift, the algorithm maintains a probabilistic structure: it generates multiple hypotheses, propagates them through time, and reweights them based on new observations—closely mirroring the structure of Bayesian and SMC inference.

---

> > ### Comment · Reviewer_gn2m · 2025-06-07
> > **Thank you!**
> >
> > Thank you for your response!

---

> > > ### Author Response · Authors · 2025-06-09
> > >
> > > Thank you once more for the constructive review and for reading our rebuttal! We will make sure to add these points in our updated draft. Please let us know if there are any additional suggestions or feedback you might have to further improve our paper and potentially increase the score.

---

### Official Review · Reviewer_dhRR · 2025-05-23

**Rating:** 6
**Confidence:** 3
**Ethics Flag:** 1

**Summary:**

This paper introduced ThoughtTracing, an inference-time algorithm to conduct Theory-of-Mind reasoning based on Bayesian Theory of Mind and Sequential Monte Carlo. This algorithm parses a text input into a trajectory, generating hypotheses and updating the hypothesis weight based on the current trajectory. After iterating through the entire trajectory, they aggregate the hypotheses on each step. Experiments across different methods and models show that ThoughtTracing improves ToM reasoning performance of the models. Further analysis reveals the differences in LLMs and LRMs.

**Questions To Authors:**

In line 139, the authors state that likelihood scoring based on six options outperforms logprob-based weighting. Are there any experiments that support this claim?

For methodological generalization, does ThoughtTracing enhance those reasoning models' performance on ToM reasoning?

**Reasons To Accept:**

* This paper introduces ThoughtTracing, an inference-time algorithm of ToM reasoning. This performance demonstrates consistent performance improvements over baseline LLMs.
* Analyses in the paper reveal insightful findings, including the difference between ToM questions and other domains (e.g., factual questions). The authors also find that the performance patterns in LLMs and LRMs are different, highlighting the further exploration of training paradigms for ToM reasoning.

**Reasons To Reject:**

Fine-grained analyses are needed.

* The evaluation relies on aggregated metrics and only conducts error analysis on MMToM-QA. Providing more case studies illustrating how hypotheses evolve over time or how weights update may give more granular insights. Additionally, unexplained anomalies, such as the QwQ 32B's performance on FANToM compared to other reasoning models, weaken the interpretability.
* The shorter reasoning traces in ThoughtTracing may reflect workflow constraints rather than efficiency. To distinguish between these possibilities, a more detailed explanation is needed.

---

> ### Author Response · Authors · 2025-06-02
>
> ### Case Studies on How Hypotheses Evolve over Time and How Weights are Updated
> Since the interleaved hypotheses are fairly long, we provide a summary of how they evolve over time.
>
> **State 1:** … In the Room B, there is a sofa, a cabinet, a desk, and a coffee table. … The Room D is furnished with a sofa, a coffee table, and a desk. … The Room C is equipped with eight cabinets, a microwave, a fridge, a dishwasher, a Room C table, and a stove. …. Jessica is situated in the Room B."
>
> **Action 1:** 'She proceeds towards the Room A, approaches the Room A cabinet, opens it, and subsequently closes it.'
>
> **Hypotheses:**
>
> - Jessica was curious about the contents of the cabinet in Room A, driven by a general exploratory behavior. (Weight: 0.33)
> - Jessica had a specific task or goal that required her to verify the contents of the cabinet in Room A, leading her to open and then close it after finding it empty. (Weight: 0.33)
> - Jessica believed that Room A might contain items of interest or value, motivating her to investigate the cabinet there. (Weight: 0.20)
> - Jessica intended to check if the cabinet in Room A was similar to the one in Room B, possibly looking for specific objects. (Weight: 0.12)
>
> **State 2:** None,
>
> **Action 2:** 'She then navigates her way to the Room C, heads towards the fourth Room C cabinet, opens it, and then shuts it.'
>
> **Hypotheses:**
>
> - Jessica was curious and exploratory; she noted the variability in cabinet contents (empty in Room A, Object K in Room C), leading her to believe contents differ across rooms. (Weight: 0.33)
> - Jessica had a specific goal, believed cabinets might hold a desired item, and methodically searched until she found Object K in Room C, reinforcing her belief in a purposeful search. (Weight: 0.33)
> - Jessica believed cabinets might contain valuable items, remained curious despite an empty cabinet in Room A, and was validated by finding Object K in Room C, (Weight: 0.20)
> - Jessica expected cabinets in other rooms to contain objects like those in Room B; after an empty cabinet in Room A, she kept searching and discovered Object K in Room C, altering her expectations. (Weight: 0.12)
>
> The first set of hypotheses is initialized from State 1 and Action 1. We update their weights using Action 1. Since the diversity of the hypotheses is above the threshold, rejuvenation is not performed. Additionally, the effective sample size is above the threshold, so resampling is not carried out.
>
> Next, we derive the second set of hypotheses by propagating from the first set. Consequently, the second set inherits the contents of its parent hypotheses. The likelihood of the second set is uniform, as all four previous hypotheses are approximately equally effective in predicting that Jessica is heading to Room C (i.e., Action 2). As a result, the weight distribution remains unchanged from the previous step. We will include these case studies in the Appendix of the updated draft.

---

> ### Author Response · Authors · 2025-06-02
>
> Thank you for your thoughtful feedback and for recognizing our contributions! We’re especially grateful for the recognition of ThoughtTracing as a novel inference-time algorithm for Theory-of-Mind reasoning that consistently improves LLM performance. We also appreciate the acknowledgment of our analyses that highlight important distinctions between ToM and factual reasoning, as well as the differing performance patterns between LLMs and reasoning models—pointing toward promising directions for future research in training paradigms.
>
> ### Full Table with Full Metrics
> We will update a full table of the full metrics in the appendix in the revised draft.
>
> ### Error Analysis on other Benchmarks
>
> #### Paraphrased-ToMi
> We observed a recurring pattern of incorrect perception predictions, similar to the issues noted in MMToM-QA. In true belief scenarios, the model sometimes incorrectly infers that the agent did not witness another agent moving an object—often due to a conservative estimation of perception (e.g., assuming the agent was not paying attention).
>
> #### FANToM
> The model tends to overestimate the target agent’s prior knowledge, occasionally assuming familiarity among characters. However, FANToM is designed such that characters are meeting for the first time, making these inferences incorrect.
>
> These cases from Paraphrased-ToMi and FANToM reflect our intentional design choice to **avoid benchmark-specific assumptions**—such as presuming agents always observe everything or have predefined relationships. We believe this is **a principled trade-off for generalizability**. That said, we acknowledge that incorporating benchmark-specific cues or more explicit assumptions in prompts could reduce these errors, and we plan to explore this direction in follow-up work.
>
> #### BigToM
> Due to the model’s near-perfect performance on BigToM, we did not conduct a detailed error analysis, as the number of errors was too limited to draw meaningful conclusions.
>
> #### QwQ 32B’s Underperformance on FANToM
> The QwQ model frequently produces false positives. For example, it often predicts that characters are aware of certain information when they are actually unaware. This pattern is consistent in both list type questions and binary yes or no questions. We will include these findings and analysis in the updated draft.
>
>
> ### Shorter Reasoning Traces in ThoughtTracing May Reflect Workflow Constraints
> The shorter reasoning traces produced by ThoughtTracing are not a result of artificial limits or workflow constraints. To clarify, the prompts used for hypothesis generation and propagation impose no length constraints (as detailed in Appendix C), and we did not impose any constraints on the total number of output tokens in our LLM API requests. The length of the reasoning traces produced by ThoughtTracing is primarily correlated with the length and complexity of the input text, as the method traces evolving mental states in alignment with the character’s trajectory throughout the input text.
>
> ### Performance Comparison of Six-option likelihood scoring over logprob-based weighting
> The performance comparison was conducted in the early stages of our project, during which we experimented with various likelihood estimation methods. Logprob-based approach yielded the weak performance, often being highly sensitive to prompt phrasing and unstable across examples. Moreover, practical limitations also influenced our decision. Since most closed-source APIs (e.g., GPT-4, Gemini) do not provide access to token-level log probabilities, broader and consistent evaluation using this method was not feasible.
>
> To provide a concrete comparison, we revisited this evaluation on MMToM-QA using Qwen 2.5. **The logprob-based weighting achieved a score of 0.42, while six-option likelihood scoring achieved 0.46**, supporting our empirical choice. We will include these additional details in the updated version of our paper.
>
>
> ### Performance Expectation of Reasoning models with ThoughtTracing Applied
> ThoughtTracing is a model-agnostic inference-time method, and the performance depends heavily on the capabilities of the underlying base model. Therefore, it is highly likely that reasoning models with ThoughtTracing applied would outperform the instruction-tuned models with ThoughtTracing, as their base performance is significantly better than instruction-tuned models. We see this as a promising future direction and plan to explore ThoughtTracing-enhanced reasoning models in follow-up work.

---

> > ### Comment · Reviewer_dhRR · 2025-06-07
> >
> > Thank you for your response. Now these are much clearer.

---

> > ### Author Response · Authors · 2025-06-09
> >
> > Thank you once more for your thoughtful review and for carefully reading our rebuttal! We are happy to hear that our responses have addressed your concerns. We will ensure that these are included in the revised version. If there is anything else we can do to further improve the paper and help increase the score, we would greatly appreciate your feedback!

---

### Decision · Program_Chairs · 2025-07-08

**Decision:**

Accept

**Comment:**

# Summary

This paper presents ThoughtTracing, a novel inference-time algorithm for improving Theory of Mind reasoning in LLMs. Drawing inspiration from Bayesian Theory of Mind and Sequential Monte Carlo (SMC) methods, ThoughtTracing models an agent's mental state as a trajectory of beliefs evolving over time. At each time step, the model generates multiple hypotheses, assigns likelihood-based weights, and updates them based on observations. The method requires no extra fine-tuning and operates purely during inference, making it flexible and efficient. Evaluations on multiple ToM benchmarks (e.g., MMToM-QA, BigToM, FANToM) show that ThoughtTracing enhances LLM performance over baselines like Chain-of-Thought. The approach also reveals differences in reasoning behavior between open-source and closed-source models, and between reasoning and instruction-tuned models.

# Strengths
- ThoughtTracing operates without training or supervision, enabling its integration across various LLMs and ToM datasets with minimal assumptions or engineering.
- The method introduces a principled hypothesis-tracking mechanism grounded in Bayesian Theory of Mind and SMC, distinguishing it from typical heuristic reasoning strategies.
- The paper provides diverse evaluations, case studies, and analyses—such as on hypothesis evolution and belief tracking—offering interpretability beyond pure performance metrics.

# Weaknesses
- While the method is claimed to be general, its application and empirical validation remain largely confined to ToM-style reasoning tasks.
- Core algorithmic steps like how initial hypotheses are constructed or paraphrased lack rigorous theoretical grounding or deeper analysis of their variability.
- The paper does not clearly quantify the token overhead or time cost of multi-hypothesis tracking, which is essential for real-world deployment or scaling.